# Protein Sensing Device with Multi-Recognition Ability Composed of Self-Organized Glycopeptide Bundle

**DOI:** 10.3390/ijms22010366

**Published:** 2020-12-31

**Authors:** Mao Arai, Tomohiro Miura, Yuriko Ito, Takatoshi Kinoshita, Masahiro Higuchi

**Affiliations:** Department of Life Science and Applied Chemistry, Graduate School of Engineering, Nagoya Institute of Technology, Gokiso-cho, Show-ku, Nagoya 4668-555, Japan; m.arai.005m@gmail.com (M.A.); t.miura.455@nitech.jp (T.M.); y.ito.678@nitech.jp (Y.I.); kinoshita.takatoshi@nitech.ac.jp (T.K.)

**Keywords:** sensing device, multi-recognition ability, protein-induced self-organization, amphiphilic glycopeptides, ion channel

## Abstract

We designed and synthesized amphiphilic glycopeptides with glucose or galactose at the *C*-terminals. We observed the protein-induced structural changes of the amphiphilic glycopeptide assembly in the lipid bilayer membrane using transmission electron microscopy (TEM) and Fourier transform infrared reflection-absorption spectra (FTIR-RAS) measurements. The glycopeptides re-arranged to form a bundle that acted as an ion channel due to the interaction among the target protein and the terminal sugar groups of the glycopeptides. The bundle in the lipid bilayer membrane was fixed on a gold-deposited quartz crystal microbalance (QCM) electrode by the membrane fusion method. The protein-induced re-arrangement of the terminal sugar groups formed a binding site that acted as a receptor, and the re-binding of the target protein to the binding site induced the closing of the channel. We monitored the detection of target proteins by the changes of the electrochemical properties of the membrane. The response current of the membrane induced by the target protein recognition was expressed by an equivalent circuit consisting of resistors and capacitors when a triangular voltage was applied. We used peanut lectin (PNA) and concanavalin A (ConA) as target proteins. The sensing membrane induced by PNA shows the specific response to PNA, and the ConA-induced membrane responded selectively to ConA. Furthermore, PNA-induced sensing membranes showed relatively low recognition ability for lectin from Ricinus Agglutinin (RCA120) and mushroom lectin (ABA), which have galactose binding sites. The protein-induced self-organization formed the spatial arrangement of the sugar chains specific to the binding site of the target protein. These findings demonstrate the possibility of fabricating a sensing device with multi-recognition ability that can recognize proteins even if the structure is unknown, by the protein-induced self-organization process.

## 1. Introduction

In recent years, diseases caused by new viruses, including COVID-19, have become frequent and serious global problems. For early detection and diagnosis of new virus infections, it is necessary to construct quick and simple high-sensitivity sensing devices. Many general sensing devices use antigen–antibody reactions of the immune system [1,2,3,4,5], as represented by the Enzyme-Linked Immuno Sorbent Assay (ELISA) method. This method requires preparation of the antibody specific to the target substance, and has the advantage of high specific recognition ability. Therefore, the devices cannot be constructed without first clarifying the structure of the target substance. It is known that the virus recognizes sugar chains on the surface of the target cell membrane and enters the cell to infect [6,7,8,9,10]. For example, influenza virus has two types of proteins: hemagglutinin (HA), which binds to sugar chains on the cell surface, and neuraminidase (NA), which cleaves the bound sugar chains. The HA binding to the sugar chains on the cell surface induces the adhesion between the virus and the cell, and fuses the virus and the cell membranes. Attempts to utilize ion channel structures in sensing systems have been reported [11,12,13,14,15,16,17]. Cornel et al. reported that an antigen sensing system composed of antibodies fixed on the half channel of gramicidin [18]. In this system, the target antigen is sensed from the decrease in ion permeability through the antigen-gramicidin embedded lipid bilayer membrane due to the blockage of the channel pore by the binding of the antigen to the antibody on the gramicidin. Furthermore, Vogel et al. fabricated a ligand-gated ion channel-composed melittin with antigen at the channel pore [19]. They detected the monoclonal antibody from the blockage of the channel pore associated with the binding of the target antibody. In these sensing systems that use an antibody (antigen)-channel complex, although the recognition of the target substrate can be converted to a simple electrochemical signal, it is still necessary to fabricate a recognition site corresponding to the target. We attempted to fabricate devices that mimic the specific recognition ability and use the channel structure to convert the recognition information into the electrochemical signals. The devices were fabricated to simultaneously have a substance recognition site and a signal transduction part in a peptide assembly formed by a self-organization process using complementary interaction between the target substance and peptides. Self-organization is a powerful tool to fabricate functional assemblies [20,21,22,23]. We previously reported that α-helical peptides with various functional groups at the terminals formed specific recognition sites for the target substrate by re-arrangement in the lipid membrane due to the complementary interaction between the target and peptide terminals [24]. In this study, an ion permeability path was formed inside the peptide assembly to convert the recognition signal of the target substrate into an electrochemical signal. In addition, we used various sugar chains as peptide-terminal functional groups to detect proteins. It is known that sugar chains have protein-recognizing ability, and sugar chains forming clusters show higher protein recognizing ability than single sugar chains [25]. We used the amphiphilic α-helical peptide with glucose or galactose to form a glycopeptide assembly with both substance recognition and signal transduction abilities. The protein sensor fabricated by our method was constructed by transcribing the spatial arrangement of the sugar chains in binding sites of the target protein, and allows the sensing of a substance with an unknown structure. Furthermore, it is effective as a convenient and rapid sensing system because it converts recognition information into simple electrochemical signals by blocking the channel pore due to protein re-binding.

## 2. Results and Discussion

### 2.1. Fabrication of Protein Sensing Systems Composed of Target Protein-Induced Self-Organized Glycopeptide Bundle

In this study, we attempted to improve the peptide bundle induced by the target substance. An ion channel structure capable of signal transduction ability was formed in the interior of the target substance-induced peptide bundle with specific recognition ability of the target. Specifically, we prepared a dipalmitoylphosphatidylcholine (DPPC) vesicle containing amphiphilic peptides with glucose (CH_3_CO-Cys-(Leu-Ser-Leu-Leu-Ser-Ser-Leu)_3_-glucose; Glu-LS) or galactose (CH_3_CO-Cys-(Leu-Ser-Leu-Leu-Ser-Ser-Leu)_3_-glucose; Gal-LS), and H-(Leu)_21_-OH (L) as a spacer peptide. Cysteines were introduced at the *N*-terminals of the amphiphilic peptides to fix the bundles on the surface of gold-deposited quartz crystals as electrodes by Au-thiol interaction. The chemical structures of the glycolpeptides and spacer peptide are shown in Figure 1.

The amphiphilic peptides with a -(Leu-Ser-Leu-Leu-Ser-Ser-Leu)_3_- sequence formed the ion channel [26]. Thus, the bundles formed in the membrane by complementary interactions between various lectins and terminal sugar chains of the amphiphilic peptides can be expected to have recognition specificity for the lectins, and transduction ability of the re-binding signal.

An outline of the fabrication of protein-sensing systems that have both the abilities of recognition diversity for the substance and signal transduction is shown in Figure 2. We previously reported [24] that the formation of the specific recognition site consisted of the peptide bundle in the lipid monolayer due to the complementary interactions between the peptide terminal groups and target substance existing in the aqueous phase beneath the monolayer. The fabrication of the protein sensing systems was carried out as follows: First, the target protein was added to the aqueous solution of the vesicle with the amphipathic glycopeptides in the bilayer membrane, and then the solution was heated above the gel–liquid crystal transition temperature (Tc). The glycopeptides diffuse laterally in the liquid crystal state membrane and re-arrange according to the spatial arrangement of the sugar chain binding site of the target protein. Next, the vesicle solution was quenched to a gel state below Tc to immobilize the structure of the glycopeptide assembly. Finally, the glycopeptide assembly in the lipid bilayer membrane was fixed on the gold-deposited electrode substrate to form sensing devices by the membrane fusion method.

### 2.2. Structure of Glycopeptide Bundle Induced by Target Protein in the Lipid Bilayer Membrane

We directly observed the morphology of the target protein-induced glycopeptide bundle in the DPPC bilayer membrane by TEM. The target protein was used peanut lectin (PNA) with specific binding site for galactose [27]. The TEM observation samples were prepared by freeze-fracture replica technique [28], and the glycopeptide bundles induced by PNA were observed as intramembranous particles. For comparison, the replica of the vesicle membrane before inducing the glycopeptide bundle with PNA was similarly prepared and TEM observation was performed. 

Figure 3 show the TEM images of the DPPC vesicle membranes containing glycopeptide bundles before and after inducing with PNA. The intramembranous particles, whose diameter were ca. 20 nm, were clearly observed only after PNA inducement (Figure 3a). It is known that bilayer membranes fracture along the hydrophobic surface by the freeze-fracture technique [29]. The structure observed in Figure 3a is considered to be internal structure of the bilayer membrane, not the added PAN existing at the membrane surface. Furthermore, the -(Leu-Ser-Leu-Leu-Ser-Ser-Leu)_3_- peptides form a channel ca. 4 nm in diameter consisting of a hexamer bundle [26]. The diameter of the observed intramembranous particles was larger than that of the -(Leu-Ser-Leu-Leu-Ser-Ser-Leu)_3_- channel and was comparable to that of the PNA. This implies that the Gal-LSs were re-arranged according to the spatial arrangement of the galactose binding sits of PNA, and spacer peptides, L, filled between the Gal-LSs to form the PNA-induced glycopeptide bundle.

### 2.3. Structure of Bilayer Membrane Containing Protein-Induced Glycopeptide Bundle on the Electrode Surface

#### 2.3.1. Conformation and Orientation of the Peptides in the Protein-Induced Glycopeptide Bundles Fixed on the Electrodes

The target protein-induced glycopeptide bundles were fixed on the surface of gold-deposited quartz crystals units by the membrane fusion method [30]. The structure and orientation of the peptides was investigated by Fourier transform infrared reflection-absorption spectra (FTIR-RAS) measurements.

Figure 4 shows the FTIR-RAS of the PNA-induced glycopeptide bundle in the DPPC bilayer membrane on the substrate. To eliminate the influence of DPPC on the spectrum, the only DPPC bilayer membrane coated substrate was also measured, and the difference in the spectrum was taken for evaluation. The spectrum of the glycopeptide bundle showed typical amide I (νC=O) and amide II (δN-H) absorption near 1650 cm^−1^ and 1540 cm^−1^, respectively, indicating the α-helical conformation [31]. The helicity of peptide in the glycopeptide bundle was estimated to be 76% from the ratio of peak intensity at 1645 cm^−1^ assigned to α-helical conformation to that of β-sheet (1620 cm^−1^) conformation [31], which was obtained by peak deconvolution of amide I band. The tilt angle of the α-helical axis of the peptide from the surface normal was estimated to be 7.4° from the ratio of the individual intensities of amide I to amide II absorption bands, *D* = *A*I/*A*II [32].

#### 2.3.2. Structure of Fixed Bilayer Membranes on the Electrodes

The structure of the fixed DPPC bilayer membranes with and without protein-induced glycopeptide bundles were investigated from the surface coverages obtained by the quartz crystal microbalance (QCM) measurements. Table 1 shows the immobilization amounts, W, and coverage of the DPPC bilayer membrane and the membranes containing the PNA- or ConA-induced glycopeptide bundle, on the QCM electrodes obtained from normalized frequency changes, ΔF, after membrane fusion. We also used ConA with specific binding sites for α-d-glucose, as the target protein [33].

The weight increase was calculated as 25.4 ng from the occupied area of the DPPC molecule-formed bilayer, i.e., 0.47 nm^2^, on the interface between the bilayer membrane and electrode, when the QCM electrode surface (the area of gold surface: 0.049 cm^2^) was completely covered with the DPPC bilayer membrane [34]. The calculated amount of fixed DPPC bilayer membrane containing the protein-induced glycopeptide bundle was obtained as follows. The molar ratio of DPPC:Glu-LS:Gla-LS:L was 500:1:1:8 on the interface between the bilayer membrane and electrode. The sectional area of the α-helical rod calculated from X-ray diffraction measurements of the α-helical peptide in solid films in which the molecules are packed hexagonally was shown to be 1.12 nm^2^ [35]. We used the sectional area as the occupied area of the peptide due to the peptides oriented vertically on the electrode. From these values, the weight of the DPPC bilayer membrane containing the protein-induction glycopeptide bundle when completely immobilized on the electrode surface was 25.1 ng. Although the coverage was slightly lower for the pure DPPC bilayer, the DPPC bilayers containing the PNA- and ConA-induced glycopeptide bundles almost completely covered the electrodes. We attributed this difference in coverage to the fact that the protein-induced peptide bundle was firmly immobilized on the gold surface by a gold-thiol reaction. This result shows that defect-free sensing membranes can be constructed by the membrane fusion method.

### 2.4. Recognition Ability of the Target Protein to the Sensing Membrane by QCM Method

The DPPC bilayer sensing membrane with PNA- or ConA-induced glycopeptide bundle was found to be flawlessly coated on the QCM electrode by the membrane fusion method. We determined binding amounts of proteins to the induced glycopeptide bundles by QCM measurements. Figure 5 shows the binding isotherms of PNA, ConA, and bovine serum albumin (BSA) to the PNA-induced (Figure 5a) or ConA-induced (Figure 5b) glycopeptide bundle. The molecular weights of PNA and ConA were 98,000 [27] and 416,000 [33], respectively, assuming that they formed tetramers. The molecular weight of BSA was 69,000. BSA was used as a control protein with no binding site for sugar chains.

All of the isotherms obtained by QCM measurement showed saturated curves. Comparing the detectability by the QCM method, the re-binding amount of the target protein used to induce the glycopeptide bundle was larger than that of other proteins in both sensing membranes. That is, PAN or ConA was preferentially bound to the glycolpeptide bundle induced by them, respectively. Quantitative analysis is shown below.

### 2.5. Signal Transduction Ability of the Target Protein Binding to the Sensing Membrane by Electrochemical Method

To increase the sensitivity of protein recognition, we investigated the signal transduction of protein re-binding information using the channel structure. Figure A1 shows the current response changes with the addition of PNA to the DPPC bilayer membrane containing a PNA-induced glycopeptide bundle on the QCM electrode when the ±5 mV triangular voltage was applied at 11.24 Hz. The response current waves were very similar to those derived according to Equation (7) based on the equivalent circuit in Figure A2. The equivalent circuit was a parallel circuit consisting of a capacitor (DPPC bilayer membrane; C_memb_), a resistor (glycopeptide bundle not bound to the target protein as an open channel; R_open_), the serial circuit of a capacitor (bound target protein on glycopeptide bundle; C_protein_), and a resistor (target protein bound to the glycopeptide bundle as a closed channel; R_close_). First, we verified whether the electrochemical properties of the DPPC bilayer membrane with a target protein-induced glycopeptide bundle could be expressed by the proposed equivalent circuit. Figure 6 shows the current responses of the sensing membrane to the addition of protein, and simulation obtained by Equation (7). The sensing membrane was used containing the PNA-induced glycopeptide bundle, and the added protein was PNA. In the simulation, the resistances, R_open_ = 6.00 × 10^4^ Ω and R_close_ = 1.73 × 10^4^ Ω, and capacitances, C_memb_ = 1.00 × 10^−6^ F and C_protein_ = 3.00 × 10^−6^ F, were held constant. Furthermore, x (=ΔI/ΔI_max_) was calculated from the ratio of the current changes, ΔI (Figure A4), between before and after the protein was added at peak voltage applied, and the saturated current changes, ΔI_max_ (Table 2); this is described below.

In the current responses of the actual measurements and simulations, the currents when the peak voltage applied were in good agreement, although there were differences at the rising and falling edges. This implies that the sensing membrane can be described by the proposed equivalent circuit. The differences at the rising and falling edges were considered to be because the time derivatives at the apex of the actually applied triangular voltage (Figure A1 below) were smaller than those of the ideal triangular wave (Figure A3).

We attempted to detect the protein using the sensing membrane using the electrochemical method. From the simulation, the value of the increase in the current before and after protein addition at the peak voltage, ΔI, was proportional to the fraction of the closed channel that is protein bound glycopeptide bundle (Figure A4). 

Figure 7 shows the relationships between the added protein concentration and the changes in the peak current responses, ΔI, of the PNA- or ConA-induced sensing membrane. It was found that the saturated isotherms were similar to binding isotherms obtained by the QCM method. The point to be characterized of the isotherm obtained by the electrochemical method was shown to have very high detection abilities compared with the isotherms obtained by the QCM method. That is, the PNA- and ConA-induced sensing membranes showed very large current responses to the PNA and ConA, respectively. This result suggests that the specific sensing of proteins was made possible by converting the protein binding signals to changes in ion permeability via the ion channel structure. 

### 2.6. Difference in Protein Recognition Ability by the QCM and Electrochemical Method

The protein detection ability of the DPPC bilayer membrane with the protein-induced glycopeptide bundle differed significantly between the measurement methods, i.e., the QCM and electrochemical methods. We attempted quantitative analysis of the binding isotherms obtained from the QCM and electrochemical measurements. The curves in Figure 5 and Figure 7 were obtained by curve-fitting the experimental values using the Langmuir binding isotherm. It can be seen that both curves are in good agreement with the experimental values. Table 2 shows the parameters used for curve-fitting, the binding constant, K, and the saturated binding amount, W_max_ or ΔI_max_. In the electrochemical measurement, the value of ΔI corresponds to the amount of the closed channel in the formed complex between the protein and glycopeptide bundle that acted as an ion channel. ΔI_max_ corresponds to the total channel amount. Therefore, we used ΔI and ΔI_max_ as the binding amount and saturated binding amount, respectively, for the Langmuir binding isotherm [36].

For the binding isotherm parameters obtained by the QCM measurements, the saturated binding amount, W_max_, for the protein used to induce the glycopeptide bundle was slightly larger, but a systematic relationship could not be seen in the binding constants, K. However, in the parameters obtained from the electrochemical method, the binding constant, K, to the target protein used to induce the glycopeptide bundle acting as a channel, was higher than that of other proteins, and the ΔI_max_ value corresponding to the saturated binding amount was also remarkably larger. In addition, the selective detection of the target protein to the DPPC bilayer membrane with the target protein-induced glycopeptide bundle was evaluated from the ratio to the saturated binding amounts of other proteins. The results are shown in Table 3. For example, the value of the selectivity of PNA to ConA, of 1.34, in the QCM method of the PNA-induced sensing membrane was obtained from W_max, PNA_/W_max, ConA_.

The selectivity was higher in the electrochemical method, and the detection sensitivity was 5 to 7 times higher than that in the QCM method. This result can be explained as follows. In the QCM method, the weight of protein bound to the sensing membrane was measured. Therefore, it is considered that the non-specific adsorption, and binding to sites without spatial coordination peculiar to the target protein, for example, a single sugar chain, was detected as a weight increase. In contrast, in the electrochemical method, the re-binding of the target protein to the protein-induced glycopeptide bundles acting as a channel was detected as a channel blockage. The non-specific adsorption, or the binding to the sites that did not have the spatial arrangement of sugar chains unique to the target protein, did not cause blockage of the channel. This appears to be why the electrochemical method enables more selective detection of the target protein than the QCM method. 

### 2.7. The Potential for Selective Protein Detection

The potential for selective protein detection using the channel blockage due to the target protein re-binding to the protein-induced glycopeptide bundle was demonstrated. This implies that the proteins with the same sugar binding sites can be selectively recognized if their 3D-structures are different. We performed detection of other proteins with galactose binding sites using the sensing membrane containing a PNA-induced glycopeptide bundle using the electrochemical method. The PNA-induced glycopeptide bundle has a spatial arrangement of galactose corresponded to its binding sites in PNA. Two proteins with galactose binding sites, ABA [37] and RCA120 [38], were used. Figure 8 shows the binding isotherms of PNA, ABA, and RCA120 to the PNA-induced glycopeptide bundle in the DPPC bilayer membrane using the electrochemical method. The parameters, K and ΔI_max_, obtained from Langmuir isotherms, are listed in Table 4. 

The isotherms of ABA and RCA120 were located below that of PNA, and the binding constant, K, and saturated binding amount, ΔI_max_, were smaller than those of PNA. Furthermore, the selectivity of PNA for ABA and RCA120 was 2 or more times greater. This result implies that the spatial arrangement of galactoses on the PNA-induced glycopeptide bundle is consistent with that of the galactose-binding sites of PNA, and differs from that of ABA and RCA120. The sensing membrane obtained can recognize the 3D structure of the binding sites in the target protein. This suggests that even proteins with binding sites for the same sugar chains could be selectively recognized if the spatial arrangement of the binding sites are different.

## 3. Materials and Methods 

### 3.1. Materials 

#### 3.1.1. Glycopeptides

Amino acid sequence of the glycopeptides, (Leu-Ser-Leu-Leu-Ser-Ser-Leu)_3_, was chosen as a channel forming element. DeGrado et al. reported the peptide had an amphiphilic α-helix conformation and formed a channel in the lipid bilayer membrane [26]. Cysteine was introduced at the *N*-terminal of the peptide to fix the glycopeptide assembly on the electrode surface by Au-thiol interaction. Peptide synthesis was carried out using 9-fluorenylmethoxycarbonyl (Fmoc) chemistry on Fmoc-Leucine loaded cross-linked ethoxylate acrylate (CLEAR)-acid resin [39]. Fmoc-amino acids and Fmoc-Leucine loaded CLEAR-acid resin were purchased from Peptide Institute, Inc. After the *N*-terminal of the peptide was protected by the acetyl group, peptide cleavage and deprotection of side chain protecting groups were performed at the same time to obtain CH_3_CO-Cys-(Leu-Ser-Leu-Leu-Ser-Ser-Leu)_3_-OH. A 21-mer of Leucine, H-(Leu)_21_-OH, was similarly synthesized as a spacer peptide. The glycopeptides with glucose or galactose at the *C*-terminal were obtained by the condensation reaction of CH_3_CO-Cys-(Leu-Ser-Leu-Leu-Ser-Ser-Leu)_3_-OH with *p*-aminophenyl-β-d-glucopyranoside or *p*-aminophenyl-β-d-galactopyranoside (Sigma-Aldrich Japan Inc., Tokyo Japan in *N*,*N’*-dimethylformamide (DMF, Wako Pure Chemical Industrials, Ltd.) using *N*,*N*’-diisopropylcarbodiimide and 1-hydroxy-7-azobenzotriazole (Watanabe Chemical Industrials, Ltd., Hiroshima, Japan) for 24 h at room temperature [40]. The chemical structure of glycopeptide with glucose (a; Glu-LS) and galactose (b; Gla-LS) are shown in Figure 2. In this figure, the structure of the spacer peptide H-(Leu)_21_-OH (c; L) is shown. From phenyl group-based absorbance, the contents of glucose and galactose at the peptide terminal were 99 and 98 mol%, respectively. 

#### 3.1.2. Target Proteins 

We used PNA (Wako Pure Chemical Industrials, Ltd., Osaka, Japan) and ConA (CALBIOCHEM, San Diego, CA, USA) as the target proteins. It is known that the PNA has specific binding sites for galactose [27], and ConA for glucose [33]. For comparison, BSA (Nacalai Tesque, Kyoto, Japan) with no binding site for sugar chains was used. Furthermore, to evaluate the specificity of the recognition ability of the sensing devices, we also used ABA (SEIKAGAKU Co., Tokyo, Japan) [37] and RCA120 (Funakoshi) [38] with galactose binding sites in the same manner as those of PNA, but different spatial arrangements of the binding sites. 

#### 3.1.3. Vesicles

Glycopeptides (Glu-LS and Gla-LS), spacer peptide (L), and DPPC (Wako Pure Chemical Industrials, Ltd., Osaka, Japan) were dissolved in DMF. These solutions were mixed and poured into a glass flask and then a thin film was formed on the interior surface of the flask by evaporating the solvent. The molar ratio of the glycopeptides and spacer peptide to DPPC was fixed at 0.01. The molar ratio of Glu-LS, Gla-LS, and L was 1:1:8. A quantity of 0.1 M 4-1-piperazineethanesulfonic acid (HEPES; Nacalai Tesque)-tris(hydroxymethyl)aminomethane (Tris; Nacalai Tesque, Kyoto, Japan) buffer solution (pH 7.2) was added to this flask, and sonicated by Branson Sonifier 250 for 10 min, under a nitrogen atmosphere at 0 °C. The final concentration of DPPC was 0.2 wt%. 

#### 3.1.4. Formation of Glycopeptide Bundle and Fix on the Electrode

We previously reported that heating a vesicle containing a peptide above its gel–liquid crystal transition temperature, Tc, increases the fluidity of the lipid bilayer membrane and allows the peptide to easily diffuse laterally through the membrane [24]. At this time, if a protein that specifically interacts with the sugar chains at the terminal of the glycopeptides is present, the glycopeptides are re-arranged in the membrane, and form specific binding site to the target protein on the glycopeptide bundle. The target protein was added to aqueous solution containing DPPC vesicle with glycopeptides and a spacer peptide in the bilayer membrane. The molar ratio of the added protein to the glycopeptides was 2. The Tc of DPPC is 41 °C [41]. The vesicle solution was incubated at 60 °C above Tc for 15 min to form a glycopeptide bundle. Then, the vesicle solution was quenched with ice water to immobilize the protein-induced glycopeptide bundle. The vesicle solution was refrigerated until used in later experiments. Immobilization of the DPPC bilayer membrane containing glycopeptide bundle on the electrode surface was performed using the fusion method [30]. The electrode was immersed in the vesicle solution with a protein-induced glycopeptide bundle and incubated for 24 h at room temperature to fix the DPPC bilayer membrane containing glycopeptide bundle on the electrode. We used a gold-deposited quartz crystal unit (QCMSC-SPP1AU, initium inc., kanagawa, Japan) as the electrode. The thiol group of cysteine at the *N*-terminal of the glycopeptide reacted with the gold on the electrode surface to firmly fix the protein-induced glycopeptide bundle. Then, for the purpose of removing target proteins, the electrodes were immersed in the buffer solution containing 1.0 × 10^−3^ M glucose or galactose, shaken for 24 h, and then washed with the buffer solution several times. The DPPC bilayer membranes containing the glycopeptide bundle immobilized on the electrode surfaces was immersed in a buffer solution and stored in a refrigerator until used for QCM measurement, and as working electrodes in the electrochemical measurement using the three-electrode method. 

### 3.2. Methods 

#### 3.2.1. TEM Observations

The shape of the protein-induced glycopeptide bundle in the DPPC bilayer membrane was directly observed with TEM using a freeze-fracture replica technique [28]. An aliquot of the DPPC vesicle solution containing the protein-induced glycopeptide bundle was placed on a thin gold plate, and this sample was rapidly plunged into slushy nitrogen (JEOL, Tokyo, Japan, EM-19510SNPD). Freeze-fracturing was carried out with a freeze-etching system (JEOL, JFDII EM-19500) at −130 °C and 5.0 × 10^−5^ mbar. To prepare the replica, platinum-carbon and pure carbon were evaporated at angles of 60° and 90°, respectively. For comparison, the replica of the vesicle was also formed before formation of the protein-induced glycopeptide bundle. TEM observations were carried out using a JEOL z-2500 electron microscope at an accelerating voltage of 200 kV. 

#### 3.2.2. FTIR-RAS Measurements

FTIR-RAS of DPPC bilayer membrane containing the protein-induced glycopeptide bundle on the gold-deposited quartz crystal unit were measured with a Jasco, FT-IR 6700S equipped with a Jasco, RAS Pro410-H reflection accessory. The incident angle was set at 85°.The RAS of the pure DPPC bilayer membrane on the gold substrate were measured. The 1900–1300 cm^−1^ region of difference spectra between the glycopeptide bundle/DPPC bilayer membranes and the pure DPPC bilayer membranes were analyzed as a sum of Gaussian/Lorentzian (9:1) composition of individual bands. The tilt angle, <*θ*>, of the α-helical axis of the glycopeptide from the surface normal was estimated from the ratio of the individual intensities of amide I to amide II absorption bands using Equation (1), proposed by Samulski et al. [32], where the angles of the transition moment (amide I and amide II) from the helix axis were supposed to be 39° and 75°, respectively, and *K* is a proportionality constant that related the intrinsic oscillator strengths of the amide I and amide II vibrational modes. The value of *K* has been calculated for a poly(L-Leu) in a KBr pellet to be 1.49.
(1)$$D=\frac{A_{I}}{A_{II}}=K\;\frac{0.5{\left(\sin\langle \theta \rangle \sin39\right)}^{2}+{\left(\cos\langle \theta \rangle \sin39\right)}^{2}}{0.5{\left(\sin\langle \theta \rangle \sin75\right)}^{2}+{\left(\cos\langle \theta \rangle \sin75\right)}^{2}}$$

#### 3.2.3. QCM Measurements

To confirm that the electrode surfaces obtained by the membrane fusion method were covered with the lipid bilayer membrane containing the protein-induced glycopeptide bundle without defects, the mass of the membrane fixed on the electrode was determined by QCM measurements. The fixed weight, W, was calculated from Equation (2) [42]:(2)$$W=-\frac{\Delta F\;\cdot A}{2F_{0}{}^{2}}\cdot \sqrt{\mu_{q}\cdot \rho_{q}}$$
where *A* is surface area of the electrode (0.049 cm^2^), *F*_0_ and Δ*F* are the resonant frequency (27 MHz) and normalized frequency change after membrane fusion, respectively, and *μ**_q_* and ρ*_q_* are shear modulus (2.95 × 10^11^ dyn/cm^2^) and density (2.65 g/cm^3^) of quartz crystal, respectively. Furthermore, we determined the amounts of re-binding target proteins on the sensing tip consisting of the DPPC bilayer membrane containing the protein-induced glycopeptide bundle from the QCM measurement. QCM measurements were performed using Initium, AFFINIX QNμ at 25 °C.

#### 3.2.4. Electrochemical Measurements

We evaluated the protein recognition abilities of the sensing membranes with a glycopeptide bundle in the DPPC bilayer membrane from the changes in the capacitance current of the membrane due to the blockage of the glycopeptide bundle pore by the re-binding of the added target protein. The measurements were performed using the 3-electrode method. We used the sensing chips consisting of the DPPC bilayer membrane containing the protein-induced glycopeptide bundle on the quartz crystal units as the working electrodes. Ag/AgCl and platinum electrodes were used as the reference and counter electrodes, respectively. A triangular voltage signal (±5 mV, 11.24 Hz) was applied to the working electrodes using a triangular wave generator (NIHON KOHDEN, SET-2100), and the response currents were recorded by a potentiostat (NIKKO KEISOKU, NPGS-2501-10nA). The electrolyte was 0.1 M HEPES-Tris buffer solution (pH 7.4) containing 0.1 M KCl. The analysis of the response currents was performed using the equivalent circuit shown in Figure A2. Here, a closed channel, that is, a glycopeptide bundle forming a complex with a target protein, is shown by a series circuit of a capacitor and a resistor. An open channel, that is, a glycopeptide bundle that does not form a complex, is shown by a resistor. A lipid bilayer membrane is shown by a capacitor. The sensing chips are represented by these parallel circuits, where the fraction of the closed channel is x, the capacitance of the protein bound to the glycopeptide bundle is $C_{protein}$, the resistance of the closed channel is $R_{close}$, the resistance of open channel is $R_{open}$, and the capacitance of the bilayer membrane is $C_{memb}$. First, we consider the current response of the closed channel, *I_C_*. The applied voltage, V, is the sum of the voltages applied to the capacitor (protein bound to the glycopeptide bundle, $C_{protein}$), $V_{1}$, and that to the resistor (closed channel, $R_{close}$), $V_{2}$. The time derivative of V is shown in Equation (3). Here, the current through the series circuit is equal for the capacitor and the resistor. That is, *I_C_* is as shown in Equation (4).
(3)$$\frac{dV}{dt}=\;\frac{dV_{1}}{dt}+\frac{dV_{2}}{dt}$$
(4)$$I_{C}=C_{protein}\cdot \frac{dV_{1}}{dt}=\frac{V_{2}}{R_{close}}$$

From Equations (4) and (5) is derived. By transforming Equation (4) and substituting it into Equation (3), a differential equation represented by Equation (5) can be obtained:(5)$$\frac{dV}{dt}=\frac{I_{C}}{C_{protein}}+R_{close}\cdot \frac{dI_{C}}{dt}$$

The voltage applied to the sensor chip is a triangular wave as shown in Figure A3, and its period is *T*. Equation (5) is solved by dividing it into singular points, that is, regions where the time derivative of the voltage, $V_{slope}$, is changed. As an initial condition, assuming that the current response at the singular point is 0, Equation (6) is obtained:(6)$$I_{c}=V_{slope}\cdot C_{PEG-metal}\left\{1-exp\left(1-\frac{t-\omega }{R_{close}\cdot C_{PEG-metal}}\right)\right\}\;0<t<\frac{T}{4};\;\omega =-\frac{T}{4},\;\;\frac{T}{4}<t<\frac{3}{4}T;\;\omega =\frac{T}{4},\;\;\frac{3}{4}T<t<T;\;\omega =\frac{3}{4}T$$

Therefore, the total current, $I=I_{C}+I_{C}^{\prime }+I_{m}$ through the equivalent circuit is given by Equation (7): (7)$$I=V_{slope}\cdot C_{PEG-metal}\left\{1-exp\left(1-\frac{t-\omega }{R_{close}\cdot C_{PEG-metal}}\right)\right\}\cdot x+\frac{V}{R_{open}}\cdot \left(1-x\right)+C_{memb}\cdot V_{slope}$$

Next, we consider the current response at *t* = *T*/4 shown in Figure A4, when the peak voltage is applied. At this point, the difference between the current, Δ*I*, before and after the target protein addition, is given Equation (8):(8)$$\Delta I=\left[V_{slope}\cdot C_{PEG-metal}\left\{1-exp\left(1-\frac{t-\omega }{R_{close}\cdot C_{PEG-metal}}\right)\right\}-\frac{V}{R_{open}}\right]\cdot x$$

The coefficient of *x* in Equation (8) is a constant value at t=T/4. In addition, because this value results when all channels are closed (x=1), ΔI is proportional to x, that, is the fraction of the closed channel. Figure A4 show a simulation of the current responses in the equivalent circuit to the fraction of the closed channel, and response current changes, ΔI, when the peak voltage is applied. 

## 4. Conclusions

In this study, we attempted to fabricate a new sensing device that mimics the functions of molecular assemblies, i.e., receptor and transducer, existing in biological membranes. The glycopeptide bundle formed by the complementary interaction among the target protein and sugar chains of the glycopeptides was used as a binding site. At this time, a channel structure as a signal transducer was simultaneously formed in the interior of the bundle. The sensing ability of the obtained devices depended significantly on the measurement method. The electrochemical method could detect the channel blockage by the re-binding of the target protein to the protein-induced glycopeptide bundle acting as a channel. In this method, the non-specific adsorption, or the binding to the sites that were not unique to the target protein, were not detected because the channel blockage did not occur in these cases. Furthermore, the sensing membranes fabricated using the self-organizing process are considered to be effective as sensing devices for substances with an unknown structure because the formation of the binding site and signal transducer is spontaneous. The method used in this study has the advantage that it is possible to create sensing chips with recognition diversity corresponding to various proteins using the same material. In addition, the sensing membranes are expected to have potential as sensing devices for unknown poisons and viruses, due to the easy and quick detection of the target substances using the electrochemical method.

## Figures and Tables

**Figure 1 ijms-22-00366-f001:**
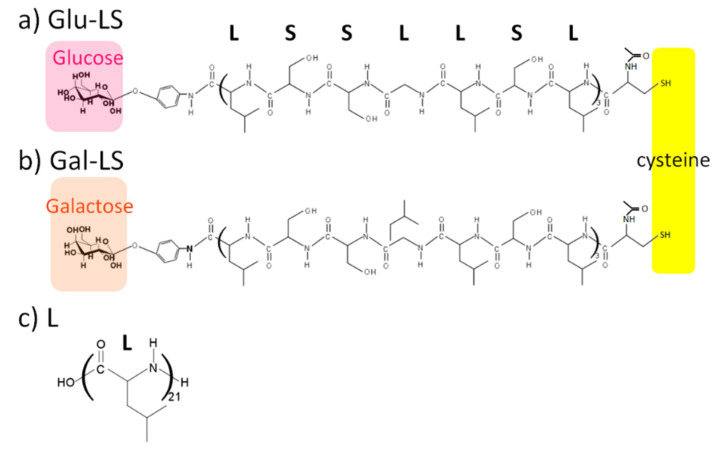
Chemical structures of glycopeptide with glucose ((**a**); Glu-LS) or galactose ((**b**); Gla-LS), and spacer peptide, H-(Leu)_21_-OH ((**c**); L). Color boxes of purple and light purple show the glucose and galactose moieties, and yellow box shows thiol group of Cysteine.

**Figure 2 ijms-22-00366-f002:**
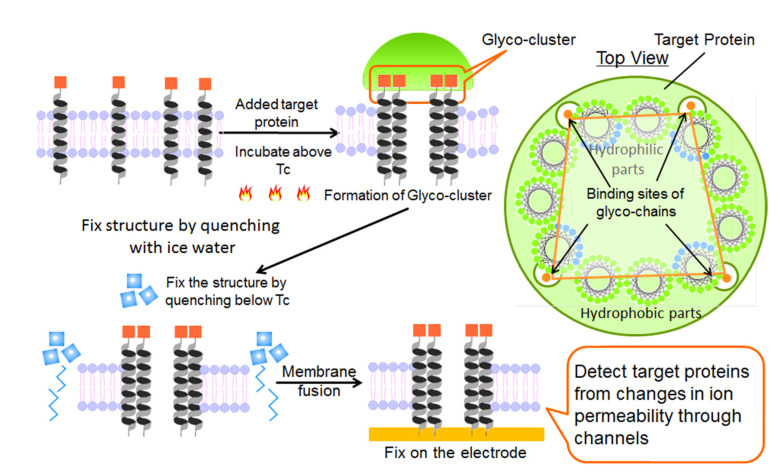
Schematic picture of fabrication for the protein-sensing systems that have both the abilities of diversity for the substance recognition and signal transduction.

**Figure 3 ijms-22-00366-f003:**
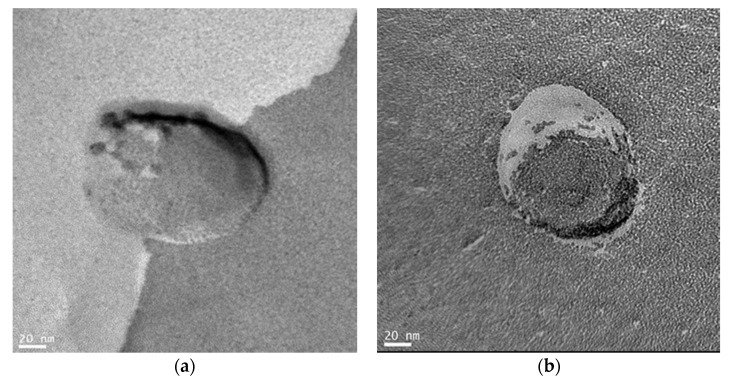
Transmission electron microscopy (TEM) images of the dipalmitoylphosphatidylcholine (DPPC) vesicle membranes containing glycopeptide bundles (**a**); after and (**b**); before inducing with peanut lectin (PNA). The scale bars show 20 nm.

**Figure 4 ijms-22-00366-f004:**
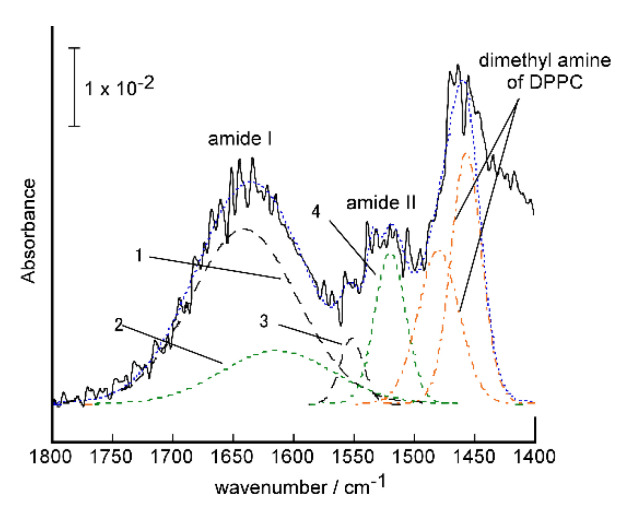
FTIR-RAS of the PNA-induced glycopeptide bundle in the DPPC bilayer membrane on the substrate, and peak deconvolution of amide I and amide II band to α-helical (1:1645 cm^−1^ and 3:1552 cm^−1^), β-sheet (2:1620 cm^−1^ and 4:1520 cm^−1^) conformation, and assigned dimethyl amine of DPPC.

**Figure 5 ijms-22-00366-f005:**
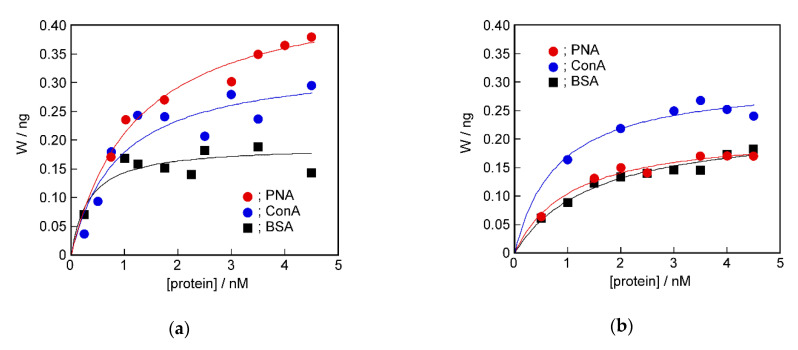
Binding isotherms of PNA (●), ConA (●), and bovine serum albumin (BSA) (■) to the DPPC bilayer membrane containing (**a**); PNA-induced and (**b**); ConA-induced glycopeptide bundle.

**Figure 6 ijms-22-00366-f006:**
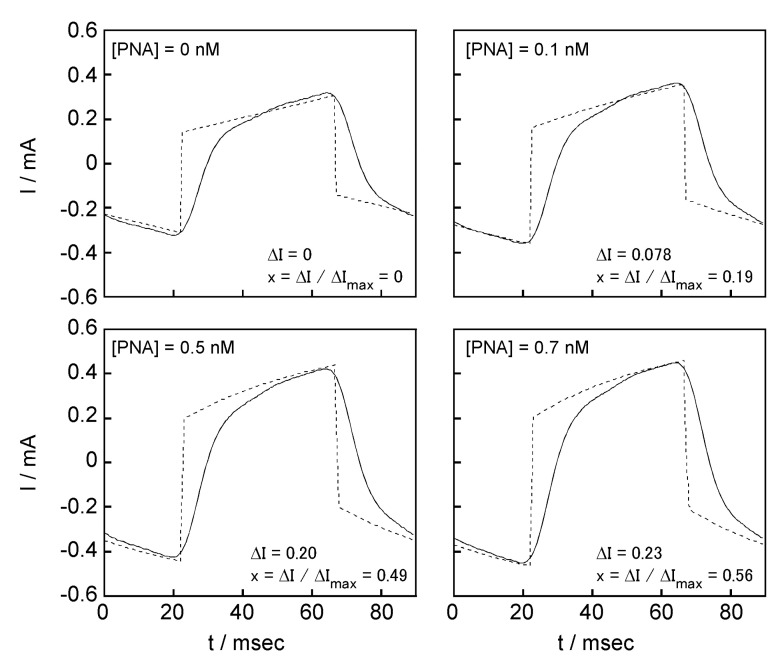
The current responses of the sensing membrane containing the PNA-induced glycopeptide bundle to the addition of PNA (solid lines), and simulation (broken lines) obtained by Equation (7).

**Figure 7 ijms-22-00366-f007:**
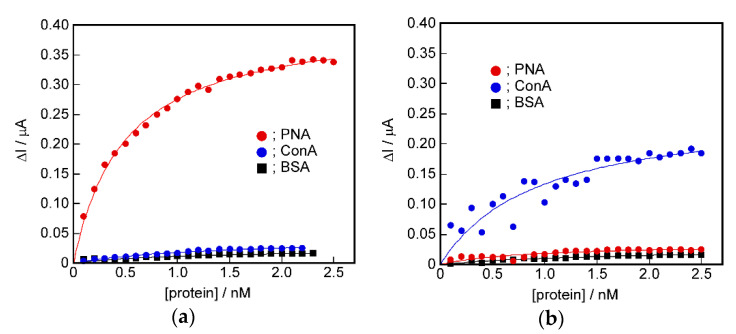
The relationships between the added protein concentration, PNA (●), ConA (●), and BSA (■), and the changes in the peak current responses, ΔI, of the DPPC bilayer membrane containing: (**a**) PNA-induced, and (**b**) ConA-induced glycopeptide bundle.

**Figure 8 ijms-22-00366-f008:**
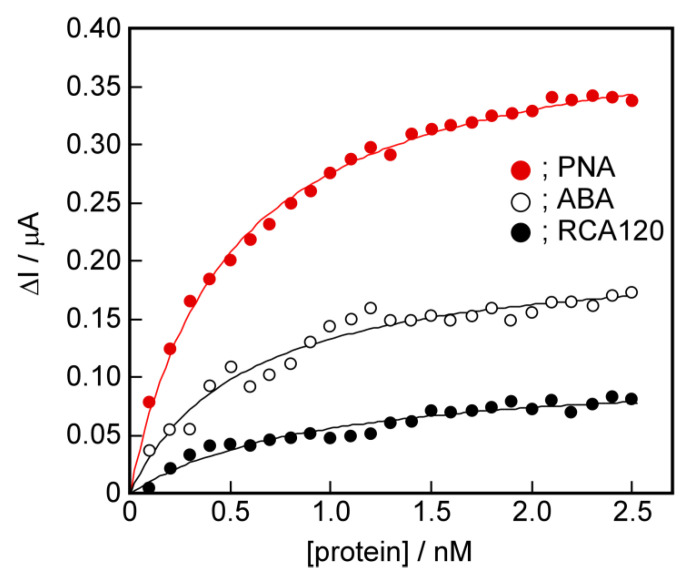
The relationships between the added protein concentration, PNA (●), ABA (〇), and RCA120 (●), and the changes in the peak current responses, ΔI, of the DPPC bilayer membrane containing a PNA-induced glycopeptide bundle. The solid lines show the Langmuir isotherms.

**Table 1 ijms-22-00366-t001:** Immobilization amounts and coverage of the DPPC bilayer membrane and the membranes containing the PNA- or ConA-induced glycopeptide bundle, on the quartz crystal microbalance (QCM) electrodes.

	ΔF/Hz	W/ng	Coverage/%
DPPC	786.0	23.6 *^1.^	91.9
PNA-induced glycopeptide bundle/DPPC	874.8	26.0 *^2.^	104
ConA-induced glycopeptide bundle/DPPC	859.4	25.5 *^2.^	102

*^1^ Immobilization amount when the surface of the QCM electrode is completely covered with DPPC bilayer membrane: 25.4 ng, and *^2^ that covered with the DPPC bilayer membrane containing PNA- or ConA-induced glycopeptide bundle: 25.1 ng.

**Table 2 ijms-22-00366-t002:** Comparison of protein detection ability using the QCM and the electrochemical methods. The sensing membranes used were DPPC bilayer membranes containing a PNA-induced or ConA-induced glycopeptide bundle. Target proteins used were PNA, ConA, and BSA.

QCM method	PNA-induced glycopeptide bundle in DPPC bilayer membrane					
	PNA		ConA		BSA	
	K/M^−1^	W_max_/ng	K/M^−1^	W_max_/ng	K/M^−1^	W_max_/ng
	8.2 × 10^8^	0.47	1.1 × 10^9^	0.34	3.1 × 10^9^	0.19
	ConA-induced glycopeptide bundle in DPPC bilayer, embrane					
	PNA		ConA		BSA	
	K/M^−1^	W_max_/ng	K/M^−1^	W_max_/ng	K/M^−1^	W_max_/ng
	9.5 × 10^8^	0.21	1.2 × 10^9^	0.31	6.7 × 10^8^	0.23
Electrochemical Method	PNA-induced glycopeptide bundle in DPPC bilayer membrane					
	PNA		ConA		BSA	
	K/M^−1^	ΔI_max_/μA	K/M^−1^	ΔI_max_/μA	K/M^−1^	ΔI_max_/μA
	2.1 × 10^9^	0.41	7.4 × 10^8^	0.043	9.5 × 10^8^	0.023
	ConA-induced glycopeptide bundle in DPPC bilayer membrane					
	PNA		ConA		BSA	
	K/M^−1^	ΔI_max_/μA	K/M^−1^	ΔI_max_/μA	K/M^−1^	ΔI_max_/μA
	1.1 × 10^9^	0.035	1.6 × 10^9^	0.26	5.8 × 10^8^	0.028

**Table 3 ijms-22-00366-t003:** Comparison of protein selectivity using the QCM and the electrochemical method. The selectivity was obtained from the saturated signal values, W_max_ and ΔI_max_, to each protein in Table 2.

QCM method	PNA-induced glycopeptide bundle in DPPC bilayer membrane	
	PNA to ConA	PNA to BSA
	1.34	2.47
	ConA-induced glycopeptide bundle in DPPC bilayer, embrane	
	ConA to PNA	ConA to BSA
	1.42	1.34
Electrochemical Method	PNA-induced glycopeptide bundle in DPPC bilayer membrane	
	PNA to ConA	PNA to BSA
	9.53	17.8
	ConA-induced glycopeptide bundle in DPPC bilayer membrane	
	ConA to PNA	ConA to BSA
	7.49	9.35

**Table 4 ijms-22-00366-t004:** Recognition characteristics of the PNA-induced glycopeptide bundle incorporated in the DPPC bilayer membrane for various proteins with galactose binding sites.

Binding Constant and Saturated Signal Value					
PNA		ABA		RCA120	
K/M^−1^	ΔI_max_/μA	K/M^−1^	ΔI_max_/μA	K/M^−1^	ΔI_max_/μA
2.1 × 10^9^	0.41	1.8 × 10^9^	0.21	1.1 × 10^9^	0.11
Selectivity of PNA to other proteins					
PNA to ABA			PNA to RCA120		
1.95			3.76		

## Data Availability

Not applicable.

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
