# Peer review of "Protein Sensing Device with Multi-Recognition Ability Composed of Self-Organized Glycopeptide Bundle"

_ijms, 2020, doi:10.3390/ijms22010366_

Round 1
Reviewer 1 Report
Comments:
In this manuscript, the authors synthesized amphiphilic glycopeptides with a glucose or a galactose at the C-terminal of each peptide (Fig. 2). It was expected that these glycopeptides could form bundles through interacting with carbohydrate-binding proteins in lipid bilayer membrane (Fig. 3). Through heating, cooling, and subsequent fixation of target protein-glycopeptides containing membrane on gold electrode (Fig. 1 ), the authors showed that the amphiphilic glycopeptides could be used to detect different proteins (Figs. 6 and 7). The results are interesting. However, due to major problems in language and data presentation, it is very difficult to read this manuscript. The manuscript requires substantial improvements in language.
Some specific comments:
- Appropriate controls should be included in the studies to support authors’ conclusions. For example peptides without glucose or galactose, modified target proteins, etc.
- 1 should be part of Results section and should not be in Introduction.
- Abbreviations, such as PNA and ConA, should be spelled out when they first appeared.

Author Response
Thank you very much for the reviewer comments. We revised the manuscript according to the comments of the reviewers.
Our alterations as a result of the comments of the reviewers are:
General comments:
Our manuscript was edited by MDPI.
Some specific comments
・As control experiments, we conducted recognition experiments using BSA that does not interact with sugar chains. From these results, we concluded that the specific binding site was formed by the interaction with the target protein. We did not conduct the experiments because the peptide without glucose or galactose does not interact specifically with specific proteins. Furthermore, instead of the experiments using the modified protein you pointed out, we conducted recognition experiments of RAC120 and ABA having galactose binding sites to the PNA-induced glycopeptide bundle. We are very interested in the experiment using the modified protein, and we would like to do it in the future.
・According to your comment, Figure 1 and its description were move to “2.2 Fabrication of protein sensing systems composed of target protein induced self-organized glycopeptide bundle” of Results and Discussion section.
・According to your comment, abbreviations were spelled out when they first appeared.
I would very much appreciate the valuable suggestions of you again.

Reviewer 2 Report
Summary:
In this study, amphiphilic glycopeptides with glucose/galactose at the C-terminals were synthesized and fixed by fusion on a gold-deposited QCM electrode. The terminal sugar groups act as receptors when an interaction with the target protein results in the glycopeptides re-arrangement to form a bundle that closes the ion channel. The detection of two target proteins PNA and ConA was characterized by monitoring the changes of the electrochemical properties of the membrane, and BSA detection was used as control. The signal was measured by both quartz crystal microbalance (QCM) measurement and electrochemical measurement by the three-electrode method. In addition, characterization was conducted by transmission electron microscopic (TEM) and Fourier transform infrared reflection-absorption spectra (FTIR-RAS). The sensing device showed multi-recognition ability by the protein-induced self-organization process.
General Comments:
The study findings are interesting and very well supported by empirical evidence. However, in order to highlight the importance of the study findings, additional effort should be invested in improving the manuscript for a clearer presentation to the reader as well as to complete the missing supportive information.
Specific Comments:
- I highly recommend that a native English speaker will thoroughly revise and correct the manuscript. Multiple errors were observed in the manuscript that must be corrected, and multiple sentences were hard to read and understand.
- Please make sure to define the abbreviations when they are first mentioned in the text, such as peanut lectin (PNA), concanavalin A (ConA), mushroom lectin (ABA), ELISA, etc.
- In this study 26 references were used to support the details discussed, while most of them are not from the last decade: 16 references are from before 2000, and only 6 references are from 2000-2010 and 4 references are from 2010-2020. Therefore, this puts a question mark on the relevance aspect of this study. More updated and recent evidence should be cited to support the relevance of the study background and findings.
- The abstract should be revised to clearly describe the following required information: study background, aim, methodology (e.g. to include characterization by TEM, FTIR-RAS, and QCM, and electrochemical signal measurement), findings (e.g. to include specific values that were measured and obtained), and conclusions.
- In the introduction section in lines 31-42 only one reference from 1995 was used to support the detailed claims. It would be better to provide additional and more recent evidence for the first paragraph of the introduction section, in order to provide the reader with updated background information on the study field.
- In the introduction section in lines 42-43: “We attempted to fabricate devices that mimic the specific recognition ability and use the channel structure to convert the recognition information into the electrochemical signals.” Should be only mentioned later in the last paragraph of the introduction. It is common and also better to first present the background of the study field and only in the last paragraph of the introduction to present the study itself.
- In the introduction section in lines 44-53, two studies were discussed from 1997 and 2001, to support the claim that “it is still necessary to fabricate a recognition site corresponding to the target”, which is not antibody-based. However, more updated and recent evidence must be discussed, at least from the latest decade, and if there are no other recent studies on this field it also should be explained.
- In the introduction section, from line 53 onwards only the study details are directly discussed, without additional knowledge on the background of the study. It is needed to provide additional background information for the clarity of the study as well as the understanding of the reader.
- In the results section, first, the design of the glycopeptides is presented in section 2.1, and then TEM structure characterization is presented in section 2.2. While the structure characterization continues to section 2.3 (2.3.1 FTIR-RAS and 2.3.2 QCM). Paraphs it will be clearer to also mention the TEM structure characterization under the same section: 2.2 (2.2.1 TEM, 2.2.2 FTIR-RAS, and 2.2.3 QCM).
- In the results section 2.4, specific values should be detailed to support the claims made, such as in lines 178-181: “Comparing the detectability by the QCM method, the re-binding amount of the target protein used to induce the glycopeptide bundle was larger than other proteins in both sensing membranes. That is to say, PAN or ConA was preferentially bound to the glycopeptide bundle induced by them, respectively.”
- In the results section lines 207-209: “The differences at the rising and falling edges were considered to be because the time derivatives at the apex of the applied triangular voltage (Figure A1 below) were smaller than that of the ideal triangular wave (Figure A3).” There might be an error in the numbers of the figures.
- In the discussion section, maybe tables 2, 3, and 4 and figure 8 would be more suitable in the results section than in the discussion section since it reports quantitative values that are accounted as the study findings. In addition, the text detailed in the discussion section is more suitable in the results section. Therefore, I would suggest considering combining the separate ‘results’ and ‘discussion’ sections into one section ‘results and discussion’.
- In the discussion section, a true discussion was not performed. It is usually required that to some extent the study findings will be compared with previous studies within the field that report either supportive or contradicting results. This is currently lacking in the manuscript, and it is important in order to highlight the importance of the study findings compare to others, as well as the impact of the findings on the current knowledge within the scientific field.
- In the materials and methods section in lines 290-413, the information is presented under the sub-heading ‘Materials’, it is better to separate the information into several paragraphs under separate sub-headings for a clearer understanding. Moreover, the information on where the chemicals were obtained or purchased from is missing, as well as the information on the devices used.
- In the materials and methods section, several references were cited, numbered: 10, 23, 24, 11, 17, 21, 22, 8, 25, 14, 12, 26, from the years: 1989, 1972, 1978, 1987, 1981, 1972, 1985, 2002, 2006, 2000, 2007, 1959, respectively. Therefore, this also puts a question mark whether the methodology that is used in the study is new.
- In the conclusions section, the most important findings of the study are missing.
- In the conflicts of interest claim: “The funders had no role in the design of the study; in the collection, analyses, or interpretation of data; in the writing of the manuscript, or in the decision to publish the results.” Because it was stated that “This research received no external funding” maybe it is better to revise it to: “The authors declared no potential conflicts of interest concerning the design of the study; the collection, analyses, or interpretation of data; the writing of the manuscript, or the decision to publish the results”.
Author Response
Thank you very much for the reviewer comments. We revised the manuscript according to the comments of the reviewers.
Our alterations as a result of the comments of the reviewers are:
- Our manuscript was edited by MDPI.
- According to your comment, abbreviations were spelled out when they first appeared.
- According to your comment, we added latest references concerning background, especially sensing devices.
- According to your comment, we revised abstract to include the background, aim, methodology, finding, and conclusions.
- According to your comment, we added latest references concerning virus recognition.
- According to your comment, we changed the order of description.
- According to your comment, we added latest references concerning sensing system using ion channel.
- Regarding this part, we changed the order of description, and only described outline of the study. Details on sensing device fabrication were described in Results and Discussion section “2.2 Fabrication of protein sensing systems composed of target protein induced self-organized glycopeptide bundle”
- In the section 2.3, we described the peptide bundle structure in the vesicle membrane by the TEM observation. On the other hand, in the section 2.4, the structure of the peptides in the lipid bilayer membrane formed by membrane fusion method was evaluated by FTIR-RAS, and the fusion membrane structure was evaluated by QCM. We believe this description is optimal because the system is completely different.
- Quantitative analysis was performed in Table 2. Here, we were only comparing the binding amounts. Because it was mentioned again when comparing that obtained by electrochemical method. We added the description “Quantitative analysis will be shown later.”
- The Figure 1A below shows the actually applied triangular wave, and Figure 3A is the ideal that. There is no mistake in the numbers in the figures. We added “actually” to the description of Figure A1 below.
12 and 13
According to your comment, we combined the separate “results” and “discussion” sections into one section “results and discussion”. The contents mentioned in the Discussion section have been incorporated into the Results and Discussion section as “2.7 Difference in protein recognition ability by the QCM and electrochemical method” and “2.8 The potential for selective protein detection”
- According to your comment, “Materials” section separated into several paragraphs under separate sub-headings. Furthermore, we added information on where the chemicals were purchased.
- The references cited in the experimental section are the most original papers on the analysis methods etc., we referred to, and they are suitable for our paper.
- According to your comment, we revised the “Conclusion” section to include the most important findings of our study.
- According to your comment, we revised the conflicts of interest claim as “The authors declared no potential conflicts of interest concerning the design of the study; the collection, analyses, or interpretation of data; the writing of the manuscript, or the decision to publish the results”
I would very much appreciate the valuable suggestions of you again.

Round 2
Reviewer 1 Report
The authors need to read through the manuscript carefully to check potential errors. There is at least one error in Abstract.
The authors have successfully addressed most of the issues I raised.
Author Response
Response for Reviewer 1
Thank you very much for the reviewer comments. We revised the manuscript according to the comments of the reviewers.
Our alterations as a result of the comments of the reviewers are:
We carefully checked mistake of the manuscript.
I would very much appreciate the valuable suggestions of you again.
Reviewer 2 Report
The authors have invested major efforts to improve the clarity of the information presented in the manuscript. They have addressed the comments that were raised. In addition, they used the language editing assistance that is offered by MDPI in order to correct and improve the level of English of the text. Minor comments can still be addressed if the authors agree as well:
- The organization of the results and discussion section may be improved. First, the two first sections describe the design (2.1) and fabrication (2.2) of the sensing system. Because no direct results are reported in these two sections, the authors may consider including this information as a part of the materials and methods section. Or if they prefer to keep it in the results and discussion section, they may consider combining both 2.1 and 2.2. into one section, because each of these two sections is also relatively short. Second, the structure analysis is currently in section 2.3 and then sub-divided. The overall organization of this section may be confusing to the reader.
- The organization of the materials and methods section may be improved. This section usually starts with a paragraph that only lists the names and details (catalog numbers and company details) of the chemicals/materials that were used in the study. Only after, in separate sub-sections, the methodologies are described. While the sub-sections names were added, the numbering of the sub-headings is still missing. The current layout may be confusing to the reader.
- A discussion is still missing from the new combined results and discussion section. It is usually required that to some extent the study findings will be compared with previous studies within the field that report either supportive or contradicting results. This is currently lacking in the manuscript, and it is important in order to highlight the importance of the study findings compare to others, as well as the impact of the findings on the current knowledge within the scientific field.
- In the conclusions section, the important results of the study are missing. The authors should consider mentioning quantitative values in the conclusions section that are the most important findings of the study.
Author Response
Response for Reviewer 2
Thank you very much for the reviewer comments. We revised the manuscript according to the comments of the reviewers.
Our alterations as a result of the comments of the reviewers are:
- Although 2.1 and 2.2 did not describe results, they described the sensor fabrication using self-organization, which was the basis of this paper, so they described in the “Results and Discussion”. According to the comment, we combined they into one session. In the section 2.2 of the revised manuscript, we described the peptide bundle structure in the vesicle membrane by the TEM observation. On the other hand, in the section 2.3 of the revised manuscript, the structure of the peptides in the lipid bilayer membrane formed by membrane fusion method was evaluated by FTIR-RAS, and the fusion membrane structure was evaluated by QCM. We believe this description is optimal because the system is completely different.
- According to the comment, we numbered the sub-heading.
- The reviewer states that “A discussion is still missing”. However, we discussed that self-organized glyco-cluster recognized the 3-D structure of the binding sites of the protein using PNA, ABA, and RCA120, which have galactose recognition ability. We think it has been fully discussed.
- The reviewer suggested that the authors should consider mentioning quantitative values in the conclusions section. However, this paper finds that the protein re-binding to the self-organized glyco-cluster, which have unique spatially arrangement of the sugar chains, closed the ion channel, resulting in high recognition ability. We have made the “Conclusion” on this point. We think it’s the good conclusion.
I would very much appreciate the valuable suggestions of you again.